# Expression, Characterisation, Homology Modelling and Molecular Docking of a Novel M17 Family Leucyl-Aminopeptidase from *Bacillus cereus* CZ

**DOI:** 10.3390/ijms242115939

**Published:** 2023-11-03

**Authors:** Jie Liu, Tangbing Cui

**Affiliations:** School of Biology and Biological Engineering, South China University of Technology, Guangzhou 510006, China; biliuj@mail.scut.edu.cn

**Keywords:** leucyl-aminopeptidase, *Bacillus cereus*, expression, homology modelling, molecular docking

## Abstract

Leucyl-aminopeptidase (LAP), an important metallopeptidase, hydrolyses amino acid residues from the N-terminus of polypeptides and proteins, acting preferentially on the peptide bond formed by N-terminus leucine. A new leucyl-aminopeptidase was found in *Bacillus cereus CZ*. Its gene (*bclap*) contained a 1485 bp ORF encoding 494 amino acids with a molecular weight of 54 kDa. The bcLAP protein was successfully expressed in *E. coli* BL21(DE3). Optimal activity is obtained at pH 9.0 and 58 °C. The bcLAP displays a moderate thermostability and an alkaline pH adaptation range. Enzymatic activity is dramatically enhanced by Ni^2+^. EDTA significantly inhibits the enzymatic activity, and bestatin and SDS also show strong inhibition. The three-dimensional model of bcLAP monomer and homohexamer is simulated byPHYRE2 server and SWISS-MODEL server. The docking of bestatin, Leu-Trp, Asp-Trp and Ala-Ala-Gly to bcLAP is performed using AutoDock4.2.5, respectively. Molecular docking results show that the residues Lys260, Asp265, Lys272, Asp283, Asp342, Glu344, Arg346, Gly372 and His437 are involved in the hydrogen bonding with the ligands and zinc ions. There may be two nucleophilic catalytic mechanisms in bcLAP, one involving His 437 or Arg346 and the other involving His437 and Arg346. The bcLAP can hydrolyse the peptide bonds in Leu-Trp, Asp-Trp and Ala-Ala-Gly.

## 1. Introduction

Aminopeptidases are a class of exopeptidases that can selectively release N-terminal amino acid residues from peptides and proteins [1]. These enzymes are found in a wide variety of sources, including animals, plants and microorganisms [2]. The earliest microbial aminopeptidase was discovered from *Aspergillus parasiticus* in the 1930s [3]; the first studies on bacterial aminopeptidases, stimulated by both basic and applied interests, were carried out in the 1960s [4]. So far, there have been various reported microbial species that produce aminopeptidases, including *Mucor*, *Penicillium*, *Aspergillus*, *Yeast*, *Monomonas*, *Lactobacillus*, *Bacillus*, *Actinomyces*, etc. The vast majority of aminopeptidases belong to the metalloenzymes family. The metalloaminopeptidases, which often contain two metal ions in their active sites, are the most numerous and homogeneous of the aminopeptidase family. Based on the three conserved active residues histidine, glutamine, lysine and/or aspartic acid located in the active sites, metalloaminopeptidases are classified into 25 families. These families are further subdivided into three groups by evolutionary clans: MF, MG and MH, respectively [5]. For many of these peptidase families, substrate hydrolysis often involves divalent metal ions at the active site, most commonly zinc ions. The MF clan of metalloaminopeptidases is characterised by the requirement for two zinc ions at the active site to drive catalysis, the most typical example being leucyl-aminopeptidase from bovine lens [6]. Leucyl-aminopeptidases belong to the M17 family of metallopeptidases [7]. They preferentially remove leucine residues from the N-terminus of proteins, peptides and synthetic substrates but have been shown to have variable activities on other amino acids [8]. In many cases, LAPs bind zinc [9], although leucyl-aminopeptidase from *Schizosaccharomyces pombe* and PepA from *Escherichia coli* are thought to bind manganese and are inhibited by zinc ions [10,11]. Leucyl-aminopeptidase has a wide range of potential applications, including the production of bioactive polypeptides [12], the removal of bitter taste from proteolytic products [13] and the deep hydrolysis of proteins [14,15]. Many microbial leucyl-aminopeptidases have been heterologously expressed and characterised. There are also many reports on the relationship between structure and function of leucyl-aminopeptidase [10,16,17]. The tertiary and quaternary structures of proteins provide valuable information about the molecular mechanism of their functions. X-ray crystallography and NMR provide high-resolution structures of proteins. However, these two methods are usually time-consuming and expensive and require large amounts of purified protein [18]. When only amino acid sequence data are available, homology modelling has become a useful tool for predicting the advanced structure of proteins. Homology modelling can be a very useful tool for researchers because the number of available microbial gene sequences has increased dramatically in recent years, while the number of experimentally determined microbial protein structures is still very low [19]. Molecular docking simulation can be used to model the interaction between a molecule and a protein. With the development and application of molecular docking, it has been widely used to study the catalytic mechanism, active sites, kinetic models of enzymes and the relationship between substrate and protein molecules [20].

In the current study, we reported on the expression, biochemical and molecular characterisation and structural prediction of a metallo-aminopeptidase, which was unambiguously identified as the translation product of the aforementioned bcLAP gene. This new member of the M17 family was first identified and purified. Homology modelling and molecular docking of bcLAP were performed, and the interactions between the residues in the active site of bcLAP and an inhibitor of three substrates were investigated.

## 2. Results and Discussion

### 2.1. Primary Structure Analysis of bcLAP

The bcLAP gene was obtained by PCR amplification from *B. cereus* CZ genomic DNA and subsequently sequenced. The sequencing result showed that the gene had an open reading frame (ORF) covering the sequence from the start codon ATG to the termination codon TAA. The ORF of the bcLAP gene consisted of 1485 nucleotides, encoding a protein of 494 amino acids with a predicted molecular weight of 54 kDa. The homology with LAP of a number of strains from *B. cereus* species, such as *Bacillus cereus* and *Bacillus thuringiensis*, is greater than 99%. Members of the *Bacillus cereus* group include species of *B. anthracis*, *B. cereus*, *B. thuringiensis*, *B. cytotoxicus*, *B. mycoides*, *B. pseudomycoides* and *B. weihenstephanensis* [21,22]. Comparison of the predicted amino acid sequence with protein sequences in the EMBL and Swiss Prot databases showed very high identity with cytosolic aminopeptidases from the *Bacillus cereus* group, such as *B. cereus* ATCC 14579 (Q816E3), *B. thuringiensis* subsp. tolworthi (A0A0G4CV43), *B. anthracis* CDC 684 (C3LC57), *B. mycoides* (A0A1X6QF78), *B. weihenstephanensis* FSL H7-687 (W4E7C7), *B. pseudomycoides* DSM 12442 (C3BRS1) and *B. cytotoxicus* DSM 22905 (A7GUC8), but lower identity with other species, e.g., 38% identity with LAP from the fungus *Coprinus cinereus* (Q8TGE4), 35% identity with LAP from *Mucor ambiguus* (A0A0C9MT45), 33% identity with the 487 amino acid sequence of *B. taurus* LAP (P00727), 32% identity with human leucyl-aminopeptidase (P28838), 36% identity with mouse LAP3 (Q9CPY7), 37% identity with mosquito LAP (W5J5J1) and 37% identity with *E. coli* cytosolic aminopeptidase (P68767), 36% identity with *Arabidopsis* LAP 2 (Q944P7) and 37% identity with *Nicotiana tabacum* LAP (U3PLK3). However, when the C-terminal domains of these different aminopeptidases, which contain their respective active sites, were considered, the sequence similarity was relatively higher (39, 43, 39, 39, 40, 39, 45, 43 and 43%, respectively). All residues associated with metal ion binding, catalysis or substrate binding were conserved in all these aminopeptidases (Figure 1), except for one hydrophilic residue (Ser431 in bcLAP), which showed conservative changes in the other species.

The Pfam database is often used to identify the family to which a protein belongs [23]. A leucyl-aminopeptidase belonging to the metallopeptidase family from *B. taurus* is the first reported M17 peptidase, which is considered to be the most representative enzyme [11]. Two significant Pfam-A (Pfam-A is curated and contains well-characterised protein domain families with high-quality alignments, which are maintained by using manually checked seed alignments and HMMs to find and align all members) matches to the amino acid sequence of bcLAP were found, including Peptidase_M17 (Cytosol aminopeptidase family, catalytic domain, 178-485aa) and Peptidase_M17_N (Cytosol aminopeptidase family, N-terminal domain, 15-143aa). The result suggested that bcLAP belongs to the M17 family. However, some LAPs belong to the M20 or M28 family [5]. M17 family peptidases have been shown to consist of two relatively independent functional domains: a unique N-terminal domain, which is much more variable among family members and is likely to play a regulatory role, and a well-conserved catalytic C-terminal domain (Figure 1). In a multiple sequence alignment of bcLAP with M17 family leucyl-aminopeptidases from bacteria, fungi, plants and mammals, the main differences were found to be in the sequence covering the N-terminal domain of bcLAP (one 10-residue insertion, one 4-residue insertion, one 3-residue insertion, one 22-residue deletion, one 6-residue deletion, one 7-residue deletion and one 2-residue deletion with respect to bovine LAP) [24]. When comparing the C-terminal domains of bcLAP and other LAPs, there were few differences; for example, there were two 1-residue insertions, one 4-residue insertion, two 1-residue deletions and one 2-residue deletion between bcLAP and ecLAP (P68767), there were five 1-residue deletions and one 1-residue insertion between bcLAP and fungal LAP (A0A0C9MT45), there were two 2-residue deletions, three 1-residue deletions, two 1-residue insertions and one 5-residue insert between bcLAP and plant LAP (Q944P7), there was one 1-residue insert, three 1-residue deletions and one 2-residue deletion between bcLAP and bovine LAP (P00727) (Figure 1). The C-terminal domains of the M17 family of leucylaminopeptidases from the *Bacillus cereus* family of strains were more conservative, with no residue deletion and no insert found when the C-terminal domains of bcLAP were compared with those of C3BRS1, A7GUC8, C3LC57, A0A0G4CV43 and W4E7C7.

The aforementioned M17 peptidases possessed the highly conserved primary structure of the C-terminal domain, which allowed us to predict the active residues and those involved in the coordination with co-catalytic metal ions of bcLAP. The active residues were predicted to be Lys272 and Arg346, and the coordinating residues to be Lys260, Asp265, Asp283, Asp342 and Glu344. M17 aminopeptidases also contain a highly conserved hydrophobic pocket that defines substrate specificity, and the pocket in bcLAP was composed of seven highly conserved residues (Met280, Asn340, Ala343, Thr369, Leu370, Gly372 and Ala460) and two non-conserved residues (Ser431 and Ser463) (Figure 1). Several conserved motifs were found in the C-terminal sequence of bcLAP, such as NTDAEGRL, F(L)VGKGI(L)TF(Y)DS(T)GG, ATLTG and V(I)ALG motifs. The presence of the NTDAEGRL motif suggests that the most likely subcellular location of the protein is the cytosol [8], indicating that bcLAP is a cytosolic enzyme. The function of the other motifs remains unclear.

### 2.2. Expression of bcLAP in E. coli

#### 2.2.1. Effects of Different Host–Vector Combinations on bcLAP Expression

The recombinant expression vector *p*EASY-E2-*bclap* was constructed and transformed into *E. coli* BL21(DE3), *E. coli* origamiTM(DE3) and *E. coli* rosetta(DE3). The recombinant expression vector *p*ET22b-*bclap* was constructed and transformed into *E. coli* BL21(DE3) and *E. coli* origamiTM(DE3). All recombinant bacteria were cultured in LB medium and induced with IPTG overnight at 28 °C and 200 rpm. The bacterial liquid was collected and then treated by sonication, proteins were extracted to determine the enzyme activity, and SDS-PAGE was performed. The result showed that *E. coli* origamiTM(DE3)-*p*EASY-E2-*bclap*, *E. coli* BL21(DE3)-*p*ET22b-*bclap* and *E. coli* origamiTM(DE3)-*p*ET 22b-*bclap* exhibited enzyme activity. SDS-PAGE of the recombinant protein revealed a protein band with an apparent Mr of approximately 54 kDa (Figure 2), consistent with the predicted molecular weight of bcLAP (53.52 kDa). Although the enzymatic functions of LAP from different organisms are similar, the number, molecular weights and types of subunits can vary considerably [2]. *S. tuberosum* LAP contains two identical 48 kDa subunits [25]. Yeast LAP [26] and *P. putida* LAP [27] are composed of twelve identical and six identical 45 kDa subunits, respectively, whereas bovine LAP [9], *S. cervi* LAP [28] and *S. pombe* LAP yspII [10] are all composed of six identical 54 kDa subunits. 

For *E. coli* BL21(DE3)-*p*EASY-E2-*bclap*, the recombinant proteins were equivalently expressed as intracellular proteins and inclusion bodies (Figure 2a), whereas for *E. coli* origamiTM(DE3)-*p*EASY-E2-*bclap*, *E. coli* rosetta(DE3)-*p*EASY-E2-*bclap* and *E. coli* origamiTM(DE3)-*p*ET22b-*bclap*, most of the recombinant proteins were expressed in the intracellular protein portion, while a small number of proteins were expressed as inclusion bodies (Figure 2b,c). However, for *E. coli* BL21(DE3)-*p*ET22b-*bclap*, the recombinant proteins were mainly present in the inclusion body form and secondarily in intracellular protein form (Figure 2d). Among all host bacteria and expression vector combinations, the combination of *E. coli* BL21(DE3) and *p*ET22b-*bclap* was the most conducive to bcLAP expression. Therefore, we selected the *E. coli* BL21(DE3) and *p*ET22b-*bclap* combination in the follow-up experiment.

#### 2.2.2. Optimisation of Expression Conditions 

The results are shown in Figure 3. The optimum temperature for enzyme activity and protein expression was found to be 28 °C (Figure 3a), which is similar to LAP from *B. stearothermophilus* [12]. At the optimal temperature, the enzyme activity reached 15.2 U/mL. When the temperature was below or above 28 °C, the enzyme activity and protein expression decreased. For the effect of biomass, the OD_600_ value for optimal enzyme activity (13.4 U/mL) and protein expression was found to be 0.8–1.0 (Figure 3b). When the IPTG concentration was 0.8 mM, the enzyme activity and protein expression reached the optimal level (Figure 3c), which was higher than the optimal IPTG concentration (0.1 mM) for the induction of *B. stearothermophilus* LAP. At this IPTG concentration, the enzyme activity reached 14.0 U/mL. The optimal induction time was found to be 8 h (Figure 3d), similar to the *B. stearothermophilus* LAP [12], and at this time, the enzyme activity was 15.8 U/mL. The results obtained led us to select 28 °C, OD_600_ 0.8–1.0, IPTG 0.8 mM and 8 h induction as the optimal conditions for *bclap* expression in *E. coli* BL21(DE3) among those tested in this work.

### 2.3. Purification of the Recombinant bcLAP 

bcLAP was purified from the crude enzyme solution extracted from *E. coli* origamiTM(DE3)-*p*ET22b-*bclap*. The purification process was summarised in Table 1, including successive ammonium sulphate precipitation, gel filtration chromatography, ion exchange chromatography and hydrophobic interaction chromatography. After a series of purification steps, the final purification fold and yield of the recombinant enzyme reached approximately 93 and 16%, respectively. The purified enzyme showed a single protein band on an SDS-PAGE gel (Figure 4).

### 2.4. Effects of Temperature and pH on Purified Aminopeptidase

According to the literature, the optimal reaction temperature of leucyl-aminopeptidase is about 50 °C, so the enzyme activity measured under this temperature condition was set to 100%, and the enzyme activity at other temperatures (35 °C, 40 °C, 45 °C, 50 °C, 55 °C, 58 °C, 60 °C, 65 °C and 70 °C) was measured, and the relative enzyme activity was calculated. As shown in Figure 5a, bcLAP had an optimal reaction temperature of 58 °C and maintained higher enzyme activity in the range of 50 °C to 65 °C. The optimal temperature of bcLAP was similar to extracellular LAP from *B. kaustophilus* CCRC 11,223 [29] and leucyl-aminopeptidase (BSAP) from *B. subtilis* Zj016 [30], which were generally found to have maximum activity at 60 °C, and rLAP II from *B. stearothermophilus* had maximum activity at 55 °C [12]. However, the optimal reaction temperature of bcLAP was lower than that of GtLAP (at 70 °C) [31], much higher than that of LAPs from some microorganisms, such as *Pseudomonas fluorescens* [32], *Streptomyces lividans* 1326 [33] and *B. subtilis* subsp. subtilis str. BSP1 [34]. To assess thermostability, the enzyme solutions were pre-incubated for 3 h at 30 °C, 40 °C, 50 °C, 60 °C, 70 °C and 80 °C, respectively, and then the enzyme activity was determined at 58 °C and the relative enzyme was calculated. Figure 5b showed that the activity of bcLAP was higher between 30 °C and 50 °C, retained more than 80% and still showed 61% residual activity at 60 °C, indicating that bcLAP had good thermal stability in the range of 30–60 °C. However, it was very sensitive to heat above 70 °C and residual activity was less than 20%. The thermostability of bcLAP was better than that of wild-type BSAP [35].

To investigate the effect of different pH, the bcLAP solutions were prepared using buffers with different pH values (3, 4, 5, 6, 7, 8, 9, 10, 11). The result showed (Figure 5c) that the enzyme activity of bcLAP reached the highest value at pH 9, and the bcLAP showed higher enzyme activity in the range of pH 8.0–10.0. At pH 11, more than 30% of the maximum activity was retained, while at pH lower than 6.0, hardly any activity was detected. bcLAP exhibited an alkaline pH optimum, which was similar to the pH optima of LAPs isolated from bacterial species [36,37] and some plants [25]. Enzyme solutions were incubated in buffers with different pH values (5, 6, 7, 8, 9, 10, 11) and then allowed to stand for 3 h. Enzyme activity was determined. The relative enzyme activity was defined as the ratio of the enzyme activity at different pH values to that at pH 9. After incubation for 3 h, the relative enzyme activity remained at 90%, 85% and 73% at pH 8, 9 and 10, respectively. However, the relative activity of bcLAP was 13–55% at pH 5–7 and 50% at pH 11 (Figure 5d). The result showed that the enzyme had good stability in the alkaline pH range, similar to the LAP from *B. subtilis* subsp. *subtilis* str. BSP1 [34]. 

### 2.5. Effect of Metal Ions and Chemical Reagents on Purified Aminopeptidase

Using Leu-NA as substrate, the optimum reaction pH of bcLAP was determined to be 9.0, and the optimum reaction temperature was 58 °C. When the activity was assayed at this pH and temperature, a significant increase in enzyme activity was found with different concentrations of Ni^2+^, reaching maximal activation (92.29-fold). Further investigation was carried out to see if other divalent metal cations could replace Ni^2+^, and the result showed that Mn^2+^, Co^2+^ and Mg^2+^ also improved aminopeptidase activity, reaching maximal activation (23.91-fold, 18.51-fold and 10.41-fold, respectively). While Zn^2+^ and Ba^2+^ also had a certain activating effect (maximum multiple 7.51-fold, 6.32-fold, respectively), and Cu^2+^ had a small activation effect (Table 2). The results indicate that bcLAP is a metal ion-dependent enzyme.

All aminopeptidases in the M17 peptidase family have two metal ions required for catalysis [35]. The metal ions are both manganese in bacterial aminopeptidase A, such as *E. coli* PepA [11], but zinc in mammalian aminopeptidases [38]. *S. pombe* LAP yspII shows a significant dependence on Mn^2+^ for maximum activity, whereas Zn^2+^ inhibits enzyme activity. The M17 family of metalloaminopeptidases show a wide range of metal ion dependence. In our study, Ni^2+^ had the strongest activating effect on bcLAP and Mn^2+^, Mg^2+^, and Co^2+^ were all potent activators of bcLAP. Similarly, the maximum activity of *S. aureus* LAP is observed in the presence of Ni^2+^, followed by Mn^2+^ and Co^2+^ [39]; *B. kaustophilus* LAP is strongly activated by Ni^2+^, with the maximum activity increased by more than 20 times, and the activation of other several divalent metal cations is in the following order of priority: Mn^2+^ > Co^2+^ > Cu^2+^ > Mg^2+^ [29]. The Ni^2+^ ion appears to have a strong activation on the catalytic activity of GtLAP, and the next is Mn^2+^, Co^2+^and Zn^2+^ ions [30]. *S. griseus* LAP, *A. proteolytica* LAP and bovine LAP typically utilise Zn^2+^ [8,24]. While other aminopeptidases have been reported to utilise Ca^2+^ [2], SGAP from *Streptomyces griseus* has been described as a calcium-activated enzyme and is approximately three-fold activated by Ca^2+^ [40]. Mn^2+^ has a strong stimulating effect on the catalytic activity of *H. pylori* LAP, Co^2+^, Ni^2+^ and Mg^2+^ have stimulating effects [41]. *H. longicornis* LAP appears to be significantly activated by the addition of Mn^2+^ in a concentration-dependent manner. The activity of *H. longicornis* LAP is increased in the presence of several other divalent metal cations with a rank order of Co^2+^ > Ni^2+^ > Mg^2+^ > Fe^2+^ > Cu^2+^ [42], and *B. stearothermophilus* LAPII is strongly stimulated by Co^2+^ (catalytic activity 26.35 fold), and stimulated by other metal ions in the following order of priority: Zn^2+^ > Li^2+^ > Mg^2+^ > Fe^2+^ > Fe^3+^ > Mn^2+^ > Ca^2+^ > Cu^2+^ > Ba^2+^ > Ni^2+^ [12].

The effects of various inhibitors on enzyme activity are shown in Table 3. EDTA, a metal ion chelator, was the most efficient inhibitor of bcLAP activity, inactivating the enzyme activity by approximately 85%. Bestatin, a specific inhibitor of metalloenzymes, showed a strong inhibition of bcLAP activity (35% residual activity), demonstrating the metal ion dependence of bcLAP activity. The hydrolytic activity of bcLAP was not sensitive to PMSF, leupeptin or pepstatin A. The effect of bestatin, PMSF, leupeptin or pepstatin A on the activity of bcLAP was similar to that of LAP from *Trypanosoma cruzi* [43]. EDTA, a metal chelator, showed the strongest inhibitory effect on the activity of bcLAP, which was reduced by approximately 85%. Similarly, the activity of SmLAP2 from *Schistosoma mansoni* is effectively inhibited by EDTA [44]. SDS is an anionic surfactant that has been shown to inactivate aminoacylase [45] and creatine kinase [46]. In this study, bcLAP activity was strongly inhibited by SDS.

### 2.6. Prediction and Analysis of the Secondary and Tertiary Structure of bcLAP

The 37% similarity between bcLAP and ecLAP from *E. coli*, a perfectly characterised enzyme, allowed us to gain some insight into the structure of bcLAP using the enzyme as a model. Based on the deduced amino acid sequence of bcLAP, the PSIPRED programme was used to predict the secondary structure of bcLAP, and it showed a very clear pattern of α- helices and β-sheets arranged alternately along the sequence, revealing that bcLAP belongs to an α/β-type protein [47], as has been shown to be the case with LAPs from *Bos taurus* and *E. coli*. The predicted secondary structure of bcLAP consisted of 45.6% α-helix, 18.4% β-sheet and 36.0% random coil, but the prediction by the Predictprotein programme showed a content of 38.9% α-helix, 15.6% β-sheet and 45.5% random coil. The obvious differences between the two different methods were found because the predicted secondary structure was only reliable when the C-terminal domain (amino acids 178–485) was taken into account, whereas the N-terminal domain (amino acids 15–143) of bcLAP and those of the LAPs used for comparison had very low sequence similarity.

The three-dimensional structure of a protein determines its cellular function. Various algorithms and computer packages are widely used to predict the structure of proteins from their amino acid sequences; the prerequisite for success is that the three-dimensional structure of a homologous protein with sequence identity has been solved previously [48]. Based on the fact that bcLAP and *E. coli* PepA share 37% sequence identity, which increases to 45% when comparing their C-terminal domains (Figure 1), and that the structure of the monomer and hexamer of *E. coli* ecLAP has been determined by X-ray crystal diffraction [9], we constructed three-dimensional models of bcLAP using the SWISS-MODEL program [49] and the PHYRE2 program [50]. The results are shown in Figure 6 and Figure 7. The bcLAP monomer consisted of two distinct domains (Figure 6a): a short N-terminal domain (residues 15–143) with highly variable folding and a longer, more conservative C-terminal domain (residues 178–485). Between the two domains was a long α-helix linker. The N-terminal domain of bcLAP consisted of five central β-strands and two α-helices on either side of the β-strands. In comparison, the N-terminal domain of ecLAP (Figure 6b) and ppLAP (Figure 6c) consisted of six central β-strands and two α-helices on either side of the β-strands. The core of the C-terminal domain of bcLAP presented a triple-layered structure with an eight-stranded beta-sheet sandwiched between 5-alpha helices on either side of the sheet. The aminopeptidase catalytic site was completely located in the C-terminal domain. The folding of the C-terminal domain of bcLAP was very similar to that of ecLAP (Figure 6b) and ppLAP (Figure 6c). 

The active site of ecLAP consists of two catalytic residues (Lys282 and Arg356) and five residues (Lys270, Asp275, Asp293, Asp352 and Glu354) that are synergistic with the two Zn^2+^ ions. The equivalents are Lys282 and Arg356 and Lys270, Asp275, Asp293, Asp352 and Glu354 in ppLAP; Lys262 and Arg336 and Lys250, Asp255, Asp273, Asp332 and Glu334 in bovine LAP. In bovine LAP, the Zn1-binding site, which binds to the carboxylates of Asp255, Asp332 and Glu334, is the structural site critical for biological activity and is specific for either Zn^2+^ or Co^2+^, and its bound ion is not easily exchanged, whereas the Zn2-binding site, which binds to the amino group of Lys250 and the carboxylates of Asp273 and Glu334, is the activable and regulatable site, and its bound ion allows exchange with different ions (e.g., Zn^2+^, Mn^2+^, Mg^2+^ and Co^2+^) [4,51,52]. Multiple sequence alignment showed that all of these residues are conserved in bcLAP (Figure 1); therefore, Lys272 and Arg346 were predicted to be two catalytic residues; Lys260, Asp265, Asp283, Asp342 and Glu344 were predicted to be metal-binding residues. In bcLAP, the Zn1-binding site is predicted to be occupied by a Zn^2+^, whereas the Zn2-binding site could be replaced by Ni^2+^, a strong activator of bcLAP.

M17 aminopeptidases also contain a highly conserved hydrophobic pocket that determines substrate specificity. The pathway of substrates into the active centre of some peptidases depends on the presence of a hydrophobic pocket located close to the surface of the monomer in bovine LAP, and its residue was found to be conserved in bcLAP. The pocket in bcLAP was composed of seven highly conserved residues (Met280, Asn340, Ala343, Thr369, Leu370, Gly372 and Ala460) and two unconserved residues (Ser431 and Ser463) (Figure 8a). At the Zn1 site, Glu344 generated interactions with the Zn atom through the α-amino group and the hydroxyl and carbonyl in the carboxyl group of the side chain, Asp342 generated interactions with the Zn atom through the carbonyl in the carboxyl group of the side chain and the carbonyl of the α-carboxyl group, Asp265 linked to the Zn atom through the hydroxyl in the carboxyl group of the side chain. The distortion in the molecular structure of Glu344 and Asp342 is necessary for the binding of the Zn atom. At the Zn2 site, Lys260 bound to the Zn atom through the amino group at the end of the side chain, Asp265 bound to the Zn atom also through the hydroxyl in the carboxyl group of the side chain, Asp283 bound to the Zn atom through the hydroxyl in the carboxyl group of the side chain. It has been reported that in bovine LAP, a single hydroxyl generated by a polarised water molecule linked two metal ions and served as a nucleophile on the substrate molecules in the catalytic process, and Lys262 played an important role in stabilising the structure of the enzyme–substrate complex. Therefore, we concluded that in bcLAP, there was also a hydroxyl between two Zn atoms, and Lys272 played a similar role to Lys262 of bovine LAP. Lys272 performed this function by forming a hydrogen bond between its amino group at the end of the side chain and the carbonyl of the α-carboxyl group. In addition, Asp265, Glu344 and Asp342 generated interactions by binding to water molecules (W1 and W2), and Asp283 formed hydrogen bonds with the amino group at the end of the side chain of Lys260 through its carboxyl group at the end of the side chain (Figure 8b). These additional interactions contributed to the spatial localisation of these residues around the Zn atoms.

Bestatin, a small molecule produced by *S. olivoreticuli*, can inhibit the activity of aminopeptidase and leucyl-aminopeptidase. Since the spatial structure of bcLAP hexamer overlapped strongly with that of ppLAP bestatin (Figure 7e), we could use ppLAP hexamer as a reference to investigate the mechanism of bestatin-bound bcLAP. The binding mode of bestatin in the active site of bcLAP showed high similarity with those of the complexes of bestatin-bound ppLAP and bestatin-bound blLAP [26,53]. The overlapping results of bcLAP and ppLAP-bestatin and the polar interactions between bcLAP and bestatin are shown in Figure 8c. In the active centre of ppLAP, most of the residues surrounding two metal atoms and bestatin are found to have equivalents in bcLAP. The comparison between Figure 8b,d shows that the most obvious difference is the replacement of the cross-linking water/hydroxide ion between Zn1 and Zn2 by the hydroxyl group of bestatin. At the Zn2 site, an additional metal coordination bond is formed by the terminal amino group of bestatin and Zn2, making the original five coordination bonds six. At the Zn1 site, a new coordination bond was formed between the amide oxygen of the peptide bond of bestatin and Zn1, while the original two coordination bonds between Asp342 and Zn1 and two of the original three coordination bonds between Glu344 and Zn1 were broken, reducing the number of coordination bonds from the original seven to four. The original hydrogen bond between Asp342 and Lys272 has been removed. The D-phenylalanine side chain of bestatin was located in the hydrophobic pocket composed of Met280, Thr369, Thr371, Gly373, Ile375 and Ala460 and interacted with these residues by van der Waals forces. Two hydrogen bonds were formed between the amide nitrogen of the peptide bond of bestatin and the main chain amide oxygen of Leu370 and between the terminal carboxylate group of bestatin and the main chain amide of Gly372. The terminal carboxylate group of bestatin also generated interactions with the main chain amides of Asp342 and Gly373. 

We have also constructed three-dimensional models of the bcLAP hexamer using the SWISS-MODEL programme, using ecLAP as a template. M17-LAPs have a highly conserved structure across different organisms and often form the structure of a hexamer with 32 symmetry and are likely to be described as a dimer of trimers; an inner cavity containing the six active sites is enclosed by a dimer of trimers [7,54]. The structure of bcLAP modelled by SWISS-MODEL had similar features with ecLAP and ppLAP (Figure 7a–c). When viewed along the three-fold molecular axis, the hexamer assumed a triangular shape with a triangular edge length of ~135 Å and a thickness of ~80 Å, which was the same as that of ecLAP and ppLAP (Figure 7b,c).

The catalytic region aggregated around the three-fold axis and mediated interactions between the subunits of each trimer and between the two trimers. At the centre of the hexamer was a large solvent cavity containing a number of hydrophobic amino acid residues such as Gly, Val, Leu, Ser, Ala, Thr and Pro, and containing the active site of the aminopeptidase. The cavity was accessible through three passages, one at the two-fold molecular axis and the other two at the interface between two N-terminal and two C-terminal domains. The N-terminal domains mediated the interactions between two trimers near the two-fold axis, resulting in the formation of the angular parts of the triangle. These structural features of bcLAP were highly consistent with those reported for pepA [9].

### 2.7. Molecular Docking of bcLAP

Molecular docking studies are helpful in elucidating the interactions between a protein and ligands at the molecular level [55]. The aminopeptidase (SGAK) has been docked to three different conformations of the Aβ peptide using AutoDock 4.2, revealing hydrogen bonding between active site residues and the Aβ peptide along with zinc ions [56]. Figure 9 shows the modelled 3D structure of bcLAP docked to bestatin and three substrates, Leu-Trp, Asp-Trp and Ala-Ala-Gly.

The substrate-binding pocket of the bcLAP monomer exhibited a “footprint-like” groove. The groove was similar in shape to two M17 leucyl-aminopeptidases from tomato [57] and *Helicobacter pylori* [56] and allowed peptide substrates to enter the groove and approach the metal-binding active site located on the sidewall on one side of the groove. Bestatin, two dipeptides and a tripeptide fit equivalently well into the substrate-binding pocket of the bcLAP monomer (Figure 9a,c,e). This suggests that bcLAP can hydrolyse a variety of peptides of different sequence and length. The conformation of bestatin generated by molecular docking had some differences from that generated by superimposing bcLAP and bestatin-bound ppLAP. However, the two conformations of bestatin showed similar interactions with the active site residues of bcLAP (Figure 7b and Figure 8d). However, in the complex of bcLAP-docked bestatin, no polar interactions were found between the amide nitrogen of the peptide bond of bestatin and Leu370, between the amino group on the side chain of D-phenylalanine of bestatin and Thr369. The amino group on the side chain of D-phenylalanine of bestatin did not form a polar contact with Zn2. The terminal carboxylate group of bestatin and His437 formed an additional hydrogen bond (Figure 9b). The free energy of binding and the inhibition constant (Ki) for the complex of bcLAP-docked bestatin were estimated to be −5.98 kcal/mol and 41.55 µM, respectively. Two dipeptides (Leu-Trp and Asp-Trp) and one tripeptide (Ala-Ala-Gly) were docked into the substrate binding groove of the active site (Figure 9c–h). The substrate binding groove was long and large enough to accommodate these three peptides at the active site of the bcLAP monomer. The substrate binding groove of bcLAP may be able to accommodate longer peptides, such as pentapeptides and hexapeptides, like the LAP-A enzyme [58]. The peptides in the binding pocket were surrounded by 9 to 10 residues that could form hydrogen bonds with Zn or the peptides. In the complex of bcLAP docked to Leu-Trp, the side-chain carboxyl group of Asp265 formed two hydrogen bonds with the α-amino group of the Leu residue and the oxygen in the peptide bond of Leu-Trp. The side-chain amino groups of Lys272 and Zn1 formed hydrogen bonds with the oxygen in the peptide bond of Leu-Trp. Two hydrogen bonds were found between the side-chain carboxyl group of Asp342 and the imino group on the indole ring of the Trp residue and between the oxygen in the peptide bond of Asp342 and the carboxyl group of the Trp residue of Leu-Trp. The side-chain amino group of Arg346 and the imino group on the imidazole ring of His437 showed hydrogen bonding to the carboxyl group of the Trp residue of Leu-Trp. The oxygen in the peptide bond of Leu 370 formed two hydrogen bonds with the imino group in the peptide bond and the carboxyl group of the Trp residue of Leu-Trp (Figure 9d). The free energy of binding and Ki for the complex of bcLAP docked to Leu-Trp were −7.07 kcal/mol and 6.53 µM, respectively. In the complex of bcLAP docked Asp-Trp, the carbonyl group in the peptide bond of Asp342, the imino group in the peptide bond of the main chain of Gly372 and the imino group on the imidazole ring of His437 formed hydrogen bonds with the side-chain carboxyl of the Asp residue of Asp-Trp. The side-chain carboxyl of Asp342 formed a hydrogen bond with the α-amino group of the Asp residue of Asp-Trp. Lys272 formed a polar contact with the carboxyl group of the Trp residue of Asp-Trp. The side-chain carboxyl of Asp263 formed a hydrogen bond with the imino group on the indole ring of the Trp residue of Asp-Trp (Figure 9f). The free energy of binding and Ki for the complex of bcLAP docked Asp-Trp were −5.83 kcal/mol and 53.41 µM. In the complex of bcLAP docked to Ala-Ala-Gly, the side-chain amino group of Lys260 and the side-chain carboxyl group of Glu344 formed hydrogen bonds with the terminal carboxyl group of Ala-Ala-Gly. The carbonyl group in the peptide bond and the guanidine group of Arg346 also formed hydrogen bonds with the terminal carboxyl group of Ala-Ala-Gly. The imino group on the imidazole ring of His437 formed a polar contact with the carbonyl group in the second peptide bond of Ala-Ala-Gly. Zn1 and the side-chain amino group of Lys272 showed polar contact with the carbonyl group in the first peptide bond of Ala-Ala-Gly. The side-chain carboxyl groups of Asp265 and Asp283 generated polar interactions with the terminal amino group of Ala-Ala-Gly (Figure 9h). The free energy of binding and Ki for the complex of bcLAP docked to Ala-Ala-Gly were −6.92 kcal/mol and 8.43 µM. Arg346 was predicted to be a nucleophilic residue in bcLAP. In the complexes of bcLAP docked Leu-Trp and Ala-Ala-Gly, Arg346 formed hydrogen bonds with the substrates, but in the complex of bcLAP docked Asp-Trp, no polar interaction between Arg346 and the substrate was found. In addition, Asp-Trp binding to the active site of bcLAP required more energy and higher concentration than Leu-Trp and Ala-Ala-Gly. This may indicate that the catalytic activity of bcLAP was weaker for Asp-Trp than for Leu-Trp and Ala-Ala-Gly. Interestingly, His437, which was located close to Arg346 in the active site of bcLAP, formed hydrogen bonds with all three substrates. We speculated that there could be two nucleophilic catalytic mechanisms in bcLAP, one involving His437 or Arg346 and the other involving His437 and Arg346. His437 was a non-conserved residue, and the equivalent was Gly in several leucyl-aminopeptidases aligned to bcLAP (Figure 1). This may be the unique feature of bcLAP. Figure 10 showed that Leu-Trp, Leu-Trp and Ala-Ala-Gly were all hydrolysed by bcLAP to the corresponding constituent amino acids. This result suggests that the enzyme can hydrolyse not only the peptide bond formed by N-terminal leucine but also the peptide bond formed by other amino acids.

## 3. Materials and Methods

### 3.1. Bacterial Strains, Vectors and Culture Conditions

*B. cereus* CZ was preserved in our laboratory. The cloning host strain *E. coli* DH5a, the cloning plasmid pMD18-T and the expression plasmid were purchased from TaKaRa (Bao Bioengineering (Dalian) Co., Ltd, China). The cloning host strain Trans1-T1, the expression host strain *E. coli* BL21 (DE3) and the expression plasmid *p*EASY-E2 were provided by Transgen (Beijing TransGen Biotech Co., Ltd., China). The expression host strain *E. coli* rosetta(DE3) and *E. coli* origamiTM(DE3) were provided by Novagen (Merck KGaA, Darmstadt, Germany).

*B. cereus* CZ was cultured in basal fermentation medium containing glucose (20 g/L), MgSO_4_ (1 g/L), yeast extract (10 g/L), bacterial peptone (10 g/L) and K_2_HPO_4_ (2 g/L). The culture conditions were 50 mL fermentation medium in 250 mL shake flask with 3% inoculum, pH 7.0, temperature 37 °C, shaking speed 200 rpm and culture time 48 h. *E. coli* strains and their recombinant strains were grown in Luria-Bertani (LB) medium at 37 °C. 

### 3.2. Reagents

Genomic DNA extraction kit, PCR purification kit, DNA gel extraction kit, IPTG, Taq polymerase, DL 5000 kb DNA marker, DL 5000 kb DNA marker, DNA supercoiled marker, 10 × loading buffer and restriction enzymes were provided by TaKaRa Bio-technology Co. Ltd. (Dalian, China). Plasmid miniprep kit was purchased from OMEGA Bio-technology. *p*EASY-E2 expression kit was purchased from Jin Li Shi Bio-technology. All other chemicals were analytical grade.

### 3.3. PCR Amplification of the LAP Gene (bclap)

Genomic DNA of *B. cereus* CZ was extracted according to the method of Genomic DNA Extractions Kit. Two oligonucleotide primers, S: ATG TTT GAA GTA CAA AAA GAA TTAG and R: TTA TTC TTC TCC AAA ACG CTC CAC, were designed according to the sequence of the leucyl-aminopeptidase gene (gi: 446410086) from the *B. cereus* B4264 genome in GenBank. *bclap* was amplified from *B. cereus* CZ genomic DNA using the S and R primers and then inserted into *p*EASY-E2 to construct the recombinant plasmid *p*EASY-E2-*bclap*. The other two primers were designed as follows: S22b: CGC GGA TCC ATG TTT GAA GTA CAA AAA GAA TTAG and R22b: ACG CGT CGA CTT CTT CTC CAA AAC GCT CCAC. *bclap* was amplified from *B. cereus* CZ genomic DNA using S22b and R22b primers and then inserted into *p*ET-22b to construct the recombinant plasmid, *p*ET-22b- *bclap*. The PCR amplification procedure was denaturation at 94 °C for 5 min, then 30 cycles (each with 30 s at 94 °C, 30 s at 60 °C, 2 min at 72 °C) and a final extension at 72 °C for 10 min. 

### 3.4. Transformation of Recombinant Plasmids and Expression of bclap 

*p*EASY-E2-*bclap* and *p*ET-22b-*bclap* were transformed into host bacteria using the following procedures. First, the competent cells were thawed on ice, 5 µL of diluted plasmid DNA was added to 50 µL of competent cells, the mixture was allowed to stand on ice for 30 min and transferred to a 2059 Falcon tube that had been cooled on ice. The mixture was then subjected to a thermal shock at 42 °C for 90 s. Finally, 500 μL of LB medium was added to the mixture and shaken at 37 °C for 60 min. A total of 100 μL of bacterial fluid was applied to the selective LB medium in a Petri dish, and the Petri dish was inverted in a 37 °C incubator overnight. 

Host bacteria included *E. coli* BL21 (DE3), *E. coli* rosetta (DE3) and *E. coli* origamiTM (DE3). IPTG was added to the medium of the recombinant *E. coli* strains to induce bcLAP expression.

### 3.5. Optimisation of Recombinant bcLAP Induction Conditions in Shake Flasks

Positive transformants were grown in 250 mL flasks containing 50 mL LB medium to determine their ability to express bcLAP. To increase the expression level of bcLAP, the effects of induction temperature, cell concentration, IPTG concentration and initial induction time on bcLAP expression were investigated in shaking flasks. The positive transformants were inoculated into 250 mL shaker flasks containing 50 mL LB medium at 200 rpm for different times (0, 4, 8, 12, 16 and 20 h). OD_600_ was monitored from 0.4 to 1.4 (0.4, 0.6, 0.8, 1.0, 1.2 and 1.4), and the different temperatures from 20 °C to 32 °C (20 °C, 24 °C, 28 °C, 32 °C) were investigated. Cells were collected and resuspended in 250 mL shaking flasks containing 25 mL LB with different concentrations of IPTG (0.2, 0.4, 0.6, 0.8, 1.0, 1.2 %, *v/v*) for 8 h. 

### 3.6. Preparation of Crude Enzyme Solution

Cultured cells were harvested by centrifugation at 10,000× *g* for 10 min, the supernatant was aspirated, and the pellet was resuspended with 50 mM Tris-HCl buffer (pH 9.0). The resulting suspension was centrifuged again at 20,000× *g* for 10 min, and the cells were collected again. The cells were resuspended in 50 mM Tris-HCl buffer (pH 9.0), and the suspension was sonicated for 30 min on ice. The lysate was centrifuged at 20,000× *g* for 20 min at 4 °C, and the supernatant (namely the soluble lysate) and precipitate were collected, respectively, and stored at 4 °C for further assays.

### 3.7. SDS-PAGE and Analysis of Enzyme Activity

The stored supernatant was collected in new tubes, and the precipitate was resuspended in 1 mL 50 mM Tris-HCl buffer (pH 9.0). This was boiled for 5 min and loaded onto SDS-PAGE gel. SDS-PAGE on a 12% (*w*/*v*) separating and 5% (*w*/*v*) stacking polyacrylamide gel was used to separate the above protein samples for 3 h at 20 mA (MiniVE GE Healthcare Bio-Sciences Corp., Philadelphia, PA, USA). Protein bands were visualised by staining with Coomassie Brilliant Blue R-250 staining buffer. The molecular weights of the proteins were calculated according to the average molecular weight of a protein marker ranging from 14.4 to 97.4 kDa (Bio-Rad Laboratories (Shanghai) Co., Ltd., China). Protein concentration was measured according to the Bradford method. 

The enzyme activity assay was performed as follows: 50 µL of enzyme solution was mixed with 600 µL of 50 mM Tris-HCl buffer (pH 9.0), the mixture was preheated at 58 °C for 5 min, and then 50 µL of 26 mM L-leucine-p-nitroanilide was added to the mixture. After incubation for 10 min at 58 °C, the reaction solution was placed in an ice bath, and the OD value at 405 nm was determined. One unit of enzyme activity was defined as the amount of enzyme hydrolysing 1 µM leucine-p-nitroanilide per minute.

### 3.8. Purification of Recombinant Aminopeptidase

All purification procedures were performed at 4 °C. The buffers used were buffer A: 50 mM Tris-HCl buffer, pH 9.0, buffer B: 50 mM Tris-HCl buffer containing 1 M NaCl, pH 9.0, buffer C: 50 mM Tris-HCl containing 1 M (NH_4_)_2_SO_4_, pH 9.0. (NH_4_)_2_SO_4_ powder was added to the crude enzyme solution to 30% saturation. The mixture was centrifuged at 20,000× *g* for 20 min. (NH_4_)_2_SO_4_ was then added to the supernatant to reach 70% saturation. The mixture was allowed to rest at 4 °C for a few hours. After centrifugation at 20,000× *g* for 20 min, the supernatant was removed, and the precipitate was dissolved in buffer A to the appropriate volume. The resulting solution was injected into a Sephadex G75 gel filtration chromatography column equilibrated with buffer A at a flow rate of 0.5 mL/min. The eluted components were collected in a series of 1 mL tubes. All fractions showing aminopeptidase activity were pooled. The sample was dialysed overnight at 4 °C against buffer A and concentrated by ultrafiltration. The resulting solution was then applied to an anion HiTrapCapto DEAE column on an AKTATMpure system (MiniVE GE Healthcare Bio-Sciences Corp., Philadelphia, PA, USA) pre-equilibrated with buffer A at a flow rate of 1 mL/min. After elution with a linear NaCl gradient from 0.2 to 1.0 M in buffer B, each fraction was assayed for aminopeptidase activity. The pooled active fractions were concentrated by ultrafiltration and applied to a phenyl HP hydrophobic column equilibrated with buffer C at a flow rate of 0.5 mL/min. The eluate fractions containing enzyme activity were collected together and concentrated by ultrafiltration.

### 3.9. Temperature and pH Characterisation of bcLAP

The enzyme activity assay was performed at pH 9.0 and in the temperature range of 35–70 °C to investigate the optimum temperature. The optimum pH was then determined by measuring the enzyme activity at the optimum temperature found and at different pH levels, from pH 3.0 to pH 11. The different pH values were obtained by using different buffers at 50 mM: Na_2_HPO_4_·12H_2_O-citric acid (pH 3.0–5.0), Na_2_HPO_4_·12H_2_O-NaH_2_PO_4_·2H_2_O (pH 6.0–7.0), Tris-HCl (pH 8.0–9.0) and glycine-NaOH (pH 9.6–11). The thermostability of recombinant bcLAP was tested by keeping the enzyme at different temperatures (30, 40, 50, 60, 70 and 80 °C) for 3 h and then measuring the enzyme activity. The pH stability was tested by mixing the enzyme with the buffers (50 mM) of different pH (5–11) at room temperature for 3 h and then measuring the enzyme activity. A control was established by keeping the enzyme at 4 °C for the same period of time before measuring the enzyme activity.

### 3.10. Effect of Metal Ions and Chemical Agents

The effect of metal ions, including Mg^2+^, Zn^2+^, Mn^2+^, Cu^2+^, Co^2+^, Fe^2+^ and Ni^2+^, was studied by preincubating the enzyme in 0.1 mM or 0.5 mM or 1 mM solutions of these ions at 4 °C for 3 h. As a control, the enzyme was maintained in Tris-HCl (pH 9.0) at the same concentrations but without these ions for 3 h at 4 °C and then assayed for activity. The chemical reagents included EDTA, PMSF, pepstain, leupeptin, bestatin and SDS. The enzyme was pre-incubated with each of the chemical reagents (at a final concentration of 1 mM) for 30 min at 4 °C, and then the enzyme activity was determined. For comparison, the enzyme in Tris-HCl (pH 9.0) at the same concentrations but without these inhibitors was kept at 4 °C for 30 min before being assayed for enzyme activity. 

### 3.11. Structure Prediction and Molecular Docking Analysis

The secondary structure of bcLAP was predicted using the PSIPRED programme (http://bioinf.cs.ucl.ac.uk/psipred/) (accessed on 10 March 2022) and the Predictprotein programme (https://www.predictprotein.org/) (accessed on 10 March 2022). The monomeric and C-terminal domain structures of bcLAP were homologously modelled using the PHYRE2 Protein Fold Recognition server (http://www.sbg.bio.ic.ac.uk/~PHYRE2) (accessed on 12 March 2022). The hexamer structure of bcLAP was homologously modelled using the SWISS-MODEL server (http://swissmodel.expasy.org/) (accessed on 12 March 2022) . All structural models were constructed based on the crystal structure of *E. coli* aminopeptidase A (PepA) (PDB code 1GYT). The reliability of the models was assessed using the SAVES v5.0 tool (https://servicesn.mbi.ucla.edu/SAVES/) (accessed on 12 March 2022) and the Molprobity programme (http://molprobity.biochem.duke.edu/index.php) (accessed on 12 March 2022) [59]. Molecular docking simulations were performed using AutoDock Tools 4.2.5 [60]. Bestatin was extracted from the bestatin complex structure of leucyl-aminopeptidase (ppLAP) from P*seudomonas putida* (PDB code 3H8G). Leu-Trp, Asp-Trp and Ala-Ala-Gly were obtained from PubChem (https://pubchem.ncbi.nlm.nih.gov/) (accessed on 25 June 2022). The reasonable and preferred docking conformation was selected according to the empirical binding free energy and cluster frequency [61]. The selected conformation was considered as the active binding conformation and used for further analysis by PyMOL 2.3.2 software. Molecular visualisation and graphing were performed using PyMOL 2.3.2 software. 

### 3.12. Thin-Layer Chromatography

A total of 100 µL of enzyme was incubated with 500 µL of peptide (5 mM) in 50 mM Tris-HC1 (pH 9.0) and 0.5 mM Ni^2+^ at 58 °C for 3 h. The enzymatic hydrolysate was ultrafiltrated through a 3 kDa ultrafiltration membrane, and the filtrate was collected. A 10 µL sample of the filtrate was applied to the thin silica gel plate (60 F254; Merck and Co. Inc., Rahway, NJ, USA), and chromatographic analysis was carried out at room temperature for approximately 1 h using n-butanol-88% formic acid–water (15:3:2, vol/vol) as the developing agent. The unhydrolysed peptides and constituent amino acids were run separately in parallel. The peptides and amino acids on the plates were visualised with a spray reagent containing 0.2% ninhydrin (*w*/*v*) in acetone and then dried at 100 °C for 10 min.

## 4. Conclusions

In the present study, we identified, for the first time, a LAP gene from the *B. cereus* strain and its active expression was successfully carried out in *E. coli* BL21(DE3). The recombinant bcLAP was produced, and its enzymatic activity and biochemical properties were characterised. The results showed that bcLAP could be classified into the M17 LAP family, and recombinant bcLAP exhibited moderate temperature adaptability and alkaline pH adaptation range, and Ni^2+^ could greatly enhance the enzyme activity. The 3D models of the predicted bcLAP monomer and homohexamer from *B. cereus* CZ were generated using the PHYRE2 server and SWISS-MODEL server, respectively. The C-terminal domain, monomer and homohexamer of bcLAP were very similar to those of ecLAP and ppLAP. Similar interactions between bcLAP and the inhibitor bestatin for bestatin-docked bcLAP and the overlay of bcLAP and bestatin-bound ppLAP. Leu-Trp, Asp-Trp and Ala-Ala-Gly were well docked into the active site of bcLAP, indicating that bcLAP can hydrolyse the peptide bond formed by other amino acids at the N-terminus in addition to the peptide bond formed by leucine at the N-terminus. Some key amino acid residues that bind to bestatin and substrates were found. There were probably two nucleophilic catalysis mechanisms in bcLAP: one involving His437 or Arg346, the other involving His437 and Arg346. The bcLAP could hydrolyse peptide bonds formed by various amino acids, including leucine.

## Figures and Tables

**Figure 1 ijms-24-15939-f001:**
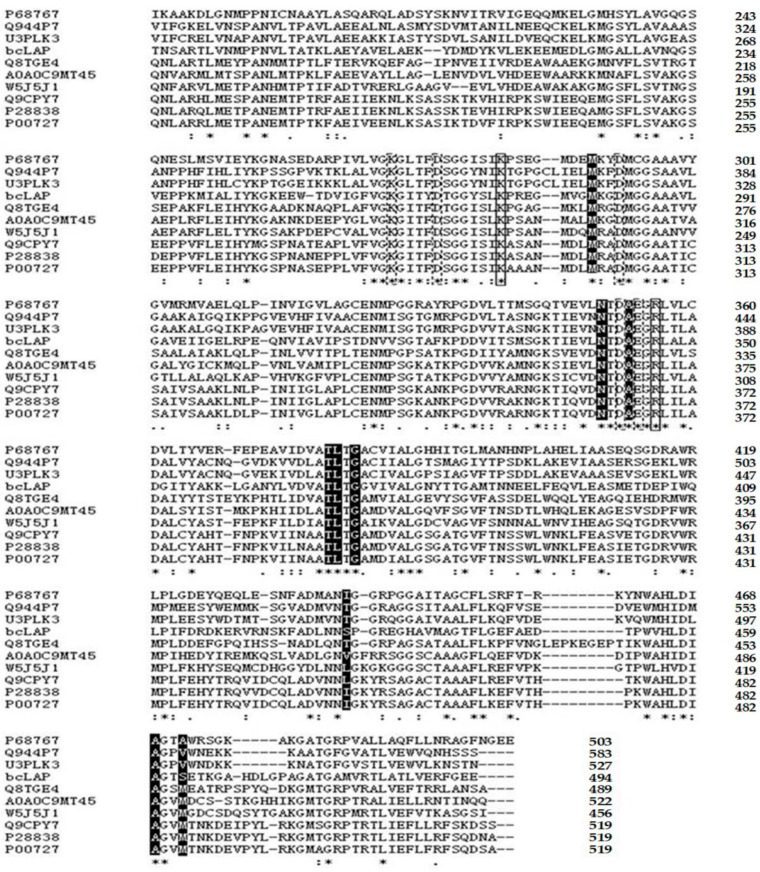
Alignment of the C-terminal domain sequences of bcLAP and leucyl-aminopeptidases from *Escherichia coli* (P68767), *Arabidopsis thaliana* (Q944P7), *Nicotiana tabacum* (U3PLK3*)*, *C. cinereus* (Q8TGE4), *Mucor ambiguous (*A0A0C9MT45), *mosquito* (W5J5J1*)*, mouse (Q9CPY7), human (P28838) and *B. taurus* (P00727). Asterisk (*): identical amino acid residues in all proteins. Solid boxes: putative catalytic residues. Dashed boxes: residues involved in coordination with the co-catalytic metal ion. Reverse-coloured residues: amino acids forming the hydrophobic pocket that defines substrate specificity.

**Figure 2 ijms-24-15939-f002:**
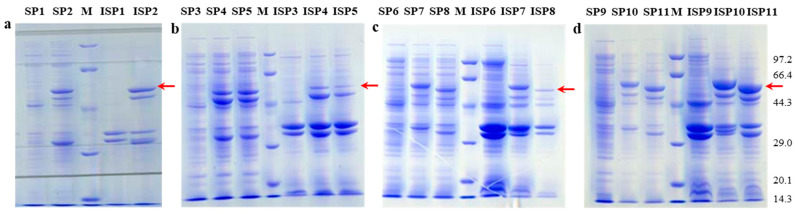
SDS-PAGE of bcLAP expression with different host and vector combinations. Red arrowheads indicated bcLAP. SP and ISP meant soluble protein and insoluble protein, respectively. M meant molecular weight marker (unit: kDa). (**a**) *E. coli* BL21(DE3) and *p*EASY-E2. SP1: the soluble protein of *E. coli* BL21(DE3); SP2: the soluble protein of *E. coli* BL21(DE3)-*p*EASY-E2-*bclap*; ISP1: the insoluble protein of *E. coli* BL21(DE3); ISP2: the insoluble body protein of *E. coli* BL21(DE3)-*p*EASY-E2-*bclap*. (**b**) *E. coli* origamiTM (DE3), *E. coli* rosetta(DE3) and *p*EASY-E2. SP3: the host soluble protein; SP4: the soluble protein of *E. coli* origamiTM(DE3)-*p*EASY-E2-*bclap*; SP5: the soluble protein of *E. coli* rosetta(DE3)-*p*EASY-E2-*bclap*; ISP3: the insoluble protein of the host bacterial; ISP4: the insoluble protein of *E. coli* origami(DE3)-*p*EASY-E2-*bclap*; ISP5: the insoluble protein of *E. coli* rosetta(DE3)-*p*EASY-E2-*bclap*. (**c**) *E. coli* origamiTM(DE3) and *p*ET22b. SP6: the soluble protein of *E. coli* origamiTM(DE3); SP7: the soluble protein of *E. coli* origamiTM(DE3)-*p*ET22b-*bclap* (with His label); SP8: the soluble protein of *E. coli* origamiTM(DE3) (without His label); ISP6: the insoluble protein of E. coli origamiTM(DE3); ISP7: the insoluble protein of *E. coli* origamiTM(DE3)-*p*ET22b-*bclap* (with His label); ISP8: the insoluble protein of E. coli origamiTM(DE3)-pET22b-*bclap* (without His label). (**d**) *E. coli* BL21(DE3) and *p*ET22b-*bclap*. SP9: the soluble protein of *E. coli* BL21(DE3); SP10: the soluble protein of *E. coli* BL21(DE3)-*p*ET22b-*bclap* (with His label); SP11: the soluble protein of *E. coli* BL21(DE3)-pET22b-*bclap* (without His label); ISP9: the insoluble protein of *E. coli* BL21(DE3); ISP10: the insoluble protein of *E. coli* BL21(DE3)-*p*ET22b-*bclap* (with His label); ISP11: the insoluble protein of *E. coli* BL21(DE3)-pET22b-*bclap* (without His label).

**Figure 3 ijms-24-15939-f003:**
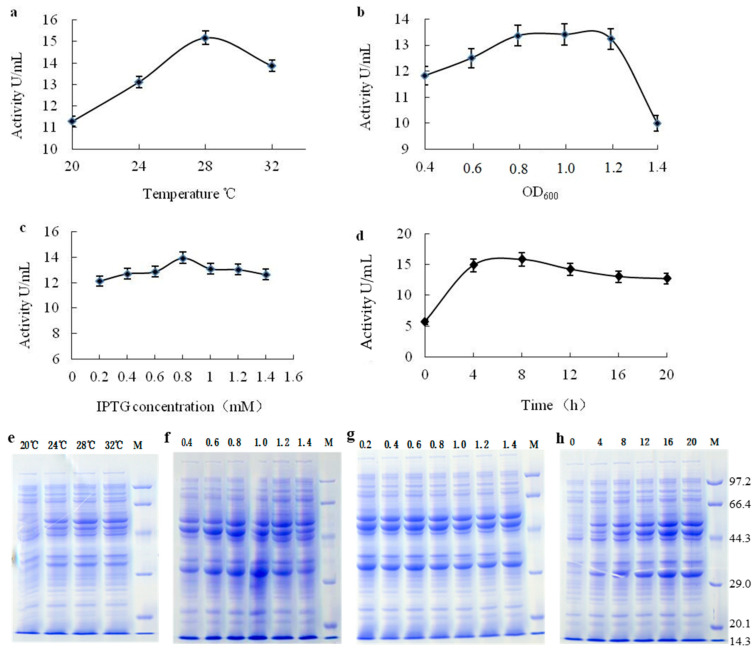
Optimisation of expression conditions for recombinant bcLAP. All samples were soluble lysate (the preparation method was shown in 3.6). (**a**) The induction temperatures (20 °C, 24 °C, 28 °C and 32 °C). (**b**) The OD_600_ values of the bacterial solution when IPTG was added. (**c**) The concentrations of IPTG. (**d**) The culture time after the addition of IPTG (final concentration was 0.8 mM) at 28 °C. (**e**) SDS-PAGE of proteins at different induction temperatures. (**f**) SDS-PAGE of proteins at different OD_600_ values of bacterial solution supplemented with 0.5 mM IPTG. (**g**) SDS-PAGE of proteins at different IPTG concentrations (mM). (**h**) SDS-PAGE of proteins at different culture times (hours). M referred to molecular weight marker (unit: kDa).

**Figure 4 ijms-24-15939-f004:**
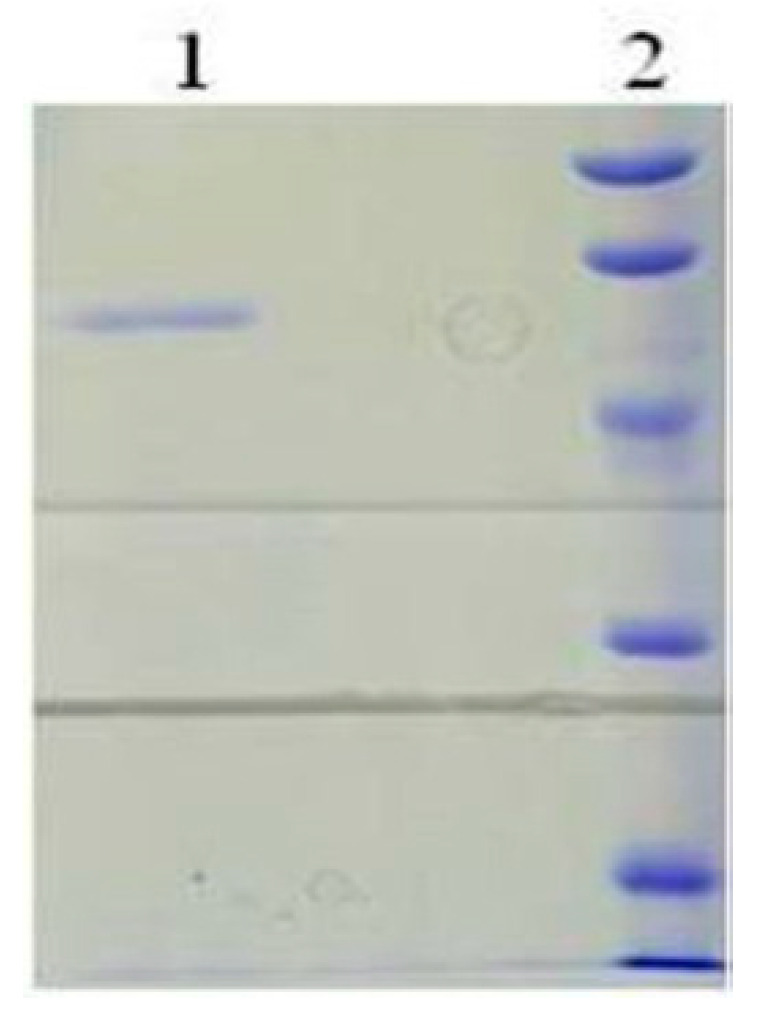
The purified bcLAP shown on an SDS-PAGE gel. Lane 1, the purified enzyme. Lane 2, the marker. The order of molecular weight markers from top to bottom of the gels was 97.2 kDa, 66.4 kDa, 44.3 kDa, 29.0 kDa, 20.1 kDa and 14.3 kDa.

**Figure 5 ijms-24-15939-f005:**
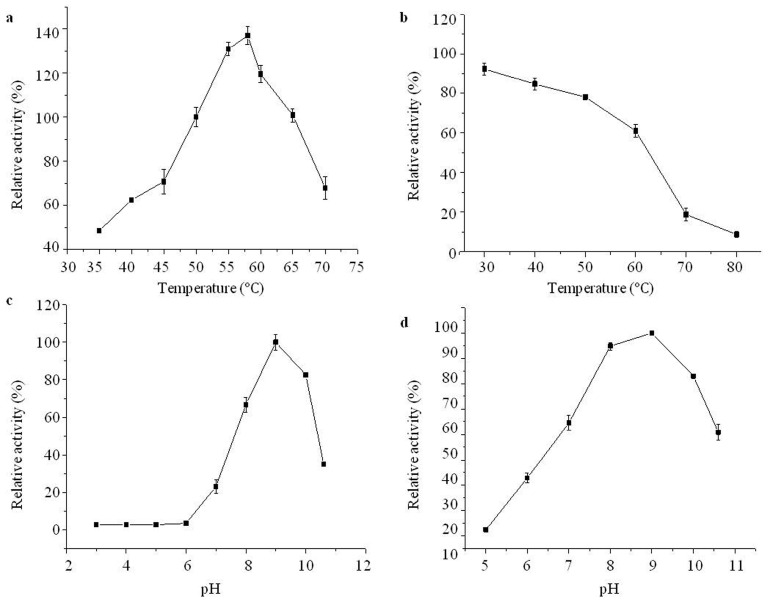
Effects of temperature and pH on the activity and stability of recombinant bcLAP. (**a**) The effect of different temperatures on enzyme activity. (**b**) Thermostability at different temperatures. (**c**) The effect of different pH on enzyme activity. (**d**) pH stability at different pH Values.

**Figure 6 ijms-24-15939-f006:**
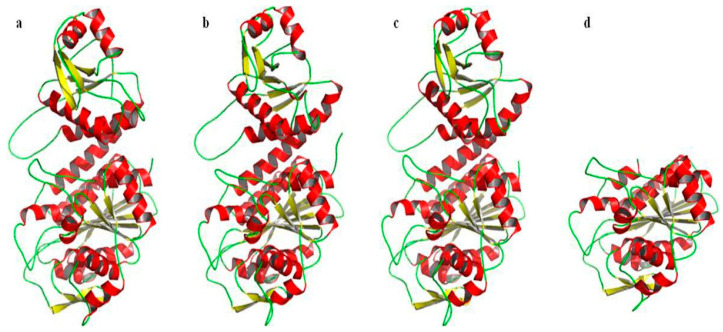
Molecular models of bcLAP. (**a**) Model of bcLAP monomer generated using the PHYRE2 server. (**b**) N-terminal (top) and C-terminal (bottom) domains of ecLAP monomer from *E. coli* by X-ray crystallography. (**c**) N-terminal (top) and C-terminal (bottom) domains of ppLAP monomer from *Pseudomonas putida* as determined by X-ray crystallography. (**d**) Model of the C-terminal domain of bcLAP generated using the PHYRE2 server.

**Figure 7 ijms-24-15939-f007:**
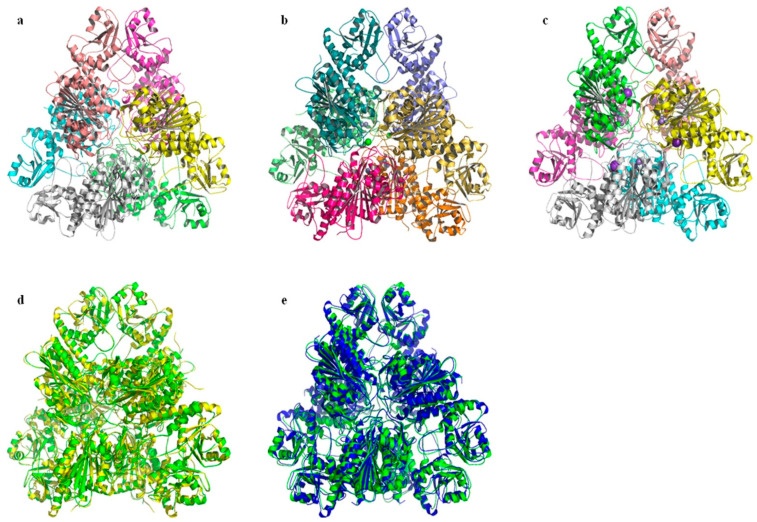
Comparison of leucyl-aminopeptidase hexamers. (**a**) bcLAP–hexamer. (**b**) ecLAP-hexamer. (**c**) ppLAP-hexamer. (**d**) bcLAP-ecLAP hexamer. (**e**) bcLAP-ppLAP hexamer.

**Figure 8 ijms-24-15939-f008:**
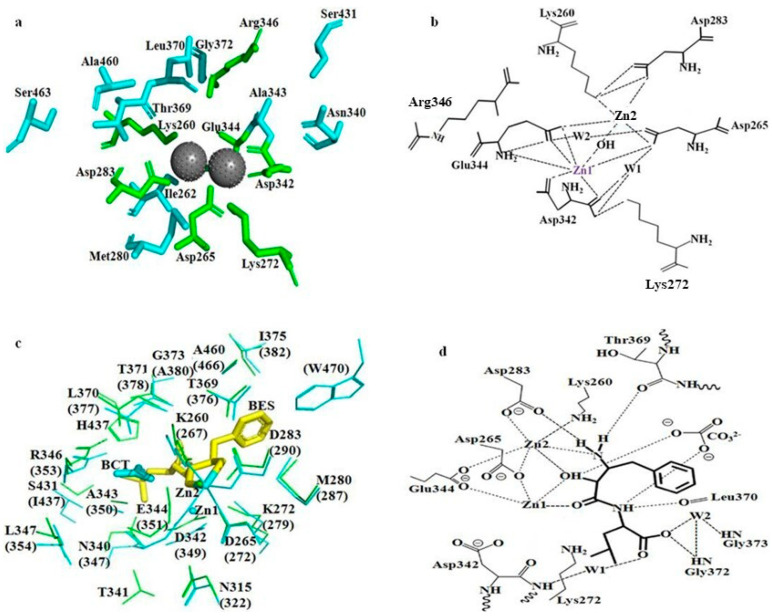
Layout of the active site of bcLAP containing two Zn atoms and a hydroxyl group. (**a**) Model visualised using PyMOL 2.3.2 software. The catalytic residues are shown in green. Residues forming the hydrophobic cavity are marked in cyan. Zn atoms are represented by a black sphere. (**b**) Schematic of hydrogen bonding interactions between Zn atoms and the catalytic residues. W1 and W2 represent water molecules. (**c**) Superposition of bcLAP and bestatin−bound ppLAP. Residues of bcLAP and residues of ppLAP are shown as green and cyan lines, respectively. Numbers in parentheses represent residue names of ppLAP. BES represents bestatin. BCT represents bicarbonate. Zn1 and Zn2 are represented by balls. (**d**) Schematic representation of the binding mode of bestatin in the active site of bcLAP. W1 and W2 represent water molecules. Hydrogen bonds and metal coordination bonds are represented by dashed lines.

**Figure 9 ijms-24-15939-f009:**
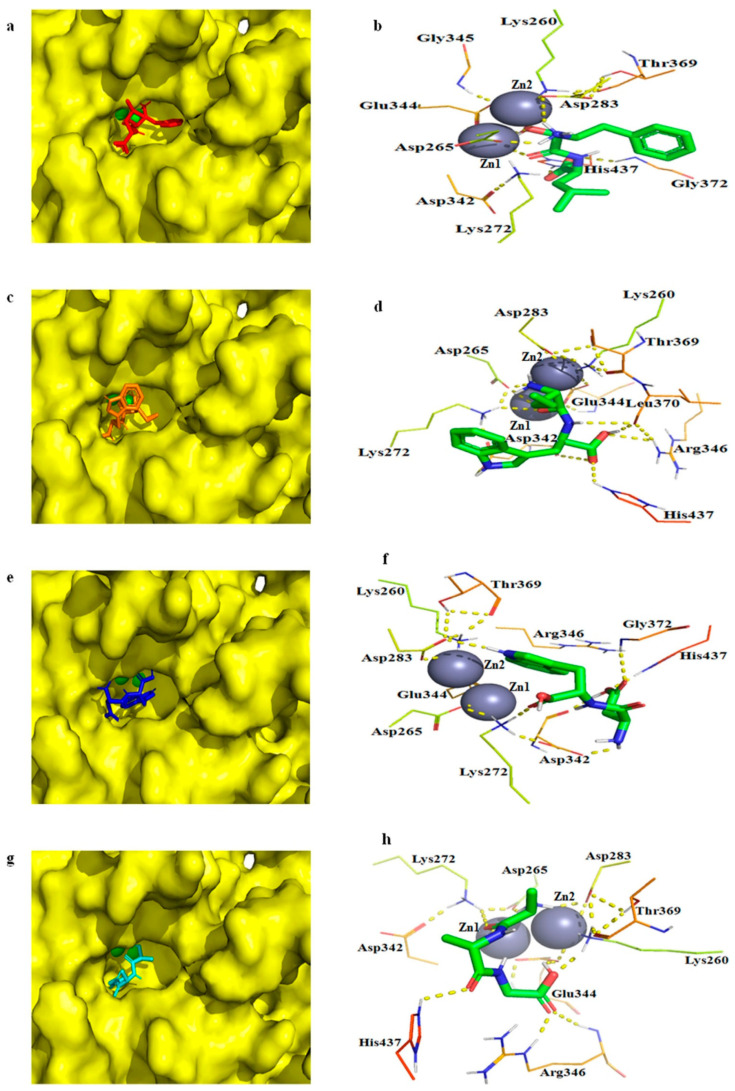
Docked complexes of the bcLAP model and interactions between active site residues with zinc and ligands. Spheres represent zinc; lines represent amino acid residues; dashes represent polar interactions; and sticks represent peptides. (**a**) The local zoom of the surface view of the complex of bcLAP docked to bestatin. (**b**) The interactions of active site residues with zinc and bestatin. (**c**) The local zoom of the surface view of bcLAP docked to Leu-Trp. (**d**) The interactions between active site residues with zinc and Leu-Trp. (**e**) The local zoom of the surface view of bcLAP docked Asp-Trp. (**f**) The interactions of active site residues with zinc and Asp-Trp. (**g**) The local zoom of the surface view of bcLAP docked Ala-Ala-Gly. Cyan bar represents Ala-Ala-Gly. (**h**) The interactions between active site residues with zinc and Ala-Ala-Gly.

**Figure 10 ijms-24-15939-f010:**
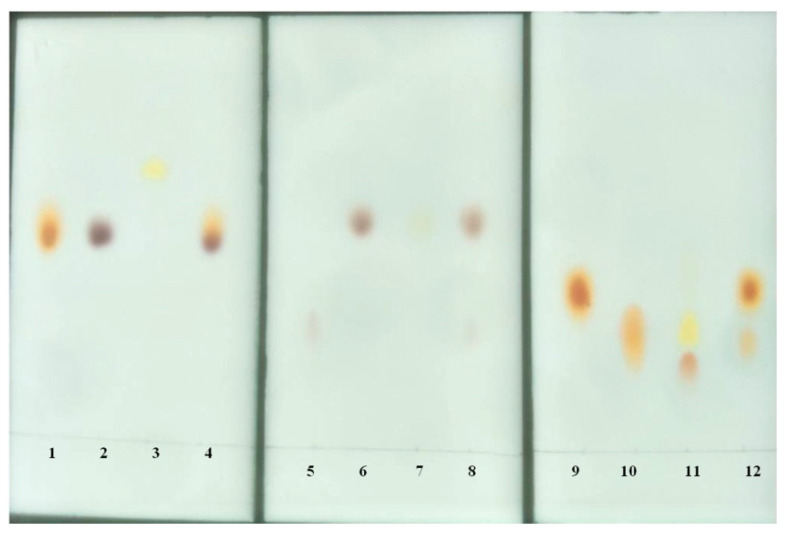
Thin-layer chromatography of peptides and constituent amino acids. Lane 1, Leu; lane 2, Trp; lane 3, Leu-Trp; lane 4, the hydrolysate of Leu-Trp; lane 5, Asp; lane 6, Trp; lane 7, Asp-Trp; lane 8, the hydrolysate of Asp-Trp; lane 9, Ala; lane 10, Gly; lane 11, Ala-Gly-Gly; lane 12, the hydrolysate of Ala-Gly-Gly.

**Table 1 ijms-24-15939-t001:** Purification of recombinant leucyl-aminopeptidase.

Purification Step	Activity (U)	Protein (mg)	Specific Activity (U/mg)	Purification (Fold)	Yield (%)
Crude extract	86,806	5695.54	15	1.00	100
Ammonium sulfate precipitation	51,950	150.40	345	22.66	59.85
Sephadex G-50	45,467	115.84	393	25.75	52.38
Capto DEAE	38,165	41.09	929	60.94	43.97
phenyl HP	14,382	10.09	1426	93.56	16.57

**Table 2 ijms-24-15939-t002:** Effect of metal ions on the activity of the recombinant bcLAP.

Concentration				Enzyme Activity Fold *			
	Ni^2+^	Mn^2+^	Co^2+^	Mg^2+^	Zn^2+^	Ba^2+^	Cu^2+^
0.1 mM	57.64 ± 0.19	23.91 ± 0.13	18.51 ± 0.08	9.22 ± 0.03	7.38 ± 0.09	6.32 ± 0.04	6.13 ± 0.03
0.5 mM	92.29 ± 0.37	21.87 ± 0.03	12.38 ± 0.07	9.55 ± 0.04	6.52 ± 0.11	6.26 ± 0.01	4.55 ±0.00
1 mM	89.60 ± 0.24	18.05 ± 0.06	11.92 ± 0.05	10.41 ± 0.10	7.51 ± 0.03	5.47 ± 0.08	2.44 ± 0.07

* Enzyme activity fold was the ratio of enzyme activity with addition of an ion to that of EDTA-treated enzyme when the enzyme activity of the EDTA-treated enzyme was 100%.

**Table 3 ijms-24-15939-t003:** Effect of chemical reagents on the activity of the recombinant bcLAP.

Chemical Reagents	Residual Activity (%) ^a^
None	100
EDTA	15.18 ± 1.21
PMSF	94.60 ± 2.02
Pepstain	97.92 ± 1.85
Leupeptin	90.45 ± 3.15
Bestatin	35.25 ± 0.89
SDS	31.68 ± 2.35

^a^ Purified leucyl-aminopeptidase (210.30 mU/mg) was preincubated in the presence of the indicated inhibitors at 58 °C for 15 min in 50 mM Tris/HCl, pH 9.0, 1 mM NiCl_2_. Enzyme activity was then determined using Leu-NA as substrate, as described in the Experimental Procedures. The concentration of the chemicals was 5 mM.

## Data Availability

The data supporting reported results can be found in this article.

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
