# Peer review of "Expression, Characterisation, Homology Modelling and Molecular Docking of a Novel M17 Family Leucyl-Aminopeptidase from Bacillus cereus CZ"

_ijms, 2023, doi:10.3390/ijms242115939_

Round 1

Reviewer 1 Report

Comments and Suggestions for Authors

General comments: This article describes, in detail, the isolation and characterisation of Bacillus cereus CZ M17 family leucyl-aminopeptidase bcLAP. The case for detailed characterisation of this enzyme and how it contributes to the canon on the leucyl peptidase family of enzymes was not effectively addressed. Can the authors clarify why they didn't pursue affinity tagging and purification of this enzyme? How did they determine the appropriate buffer conditions - 50 mM Tris seems a minimal buffer system to support solubility. Here, the authors have used Phyre2 and SWISS-MODEL to generate models of bcLAP. Did they compare these results to AlphaFold2? Given the documented superior performance of AlphaFold2 and AlphaFold multimer, it would be beneficial to compare their results with this prediction algorithm. Given the predictions of binding Zn2+ at Zn1-binding site, and Ni2+ at Zn2-binding site, the experiment described in Table 2 should be repeated with a combination of Ni and Zn. This experiment should also be repeated on EDTA treated bcLAP, see detailed comments below, including recommendations for additional experiments to be included in this manuscript. I have recommended changes to the language in some sentences to clarify their meaning where necessary.

Detailed comments:

PHYRE2, SWISS-MODEL and Molprobity should be cited in the references.

Check amino acid nomenclature throughout - three letter codes with and without spaces between the letters and numbers appear frequently.

P1 line 10 - N-terminus of polypeptides

P1 line 11 N-terminal leucine

P1 line 19: delete ', respectively'

P1 line 23: Leu-Trp is repeated. Should there be another pair listed here?

P2 line 47: delete the space in 'l eucyl-ami-'

P3 - Fig 1 - this alignment is pixellated and stretched in the x-axis dimension. Renumber the sequences to reflect the numbering of the C-terminal domains aligned in this figure. e.g. P688767 IKAAK starts at residue 184.

P3 line 5 - italicise B. taurus

P3 line 107 - define Pfam A and use correct nomenclature Pfam-A; cite the database Finn et al, NAR, 2010.

P3 line 110 - reword. Suggest replacing 'expressed' - 'The result suggested'

P4 line 112 - this reads like filler - 'and so on' is not scientific. Why are you describing these similar enzymes? And what are the 'so on' and the value of mentioning them? Either reword to define why you're going into this detail, or remove it.

P4 line 152 - reword 'The bacterial liquid was collected and then treated by sonication, proteins were extracted to determine the enzyme activity and SDS-PAGE was performed'. Bacterial liquid is not definitive. Proteins are not extracted to determine enzyme activity.

P4 line 160 - S. Tuberosum - italicise

P5 line 163 - italicise S. Cervi

P5 figure 2 - what is 'intracellular protein'? Is this the total protein extract? Or soluble extract in what buffer? Label the lanes instead of numbering them. This makes it very hard to refer to the figure legend and understand what each gel is showing. How are you doing the inclusion body extraction? This is not described in the methods. The molecular weights are defined as kD. This is not equivalent to kilodaltons. Please amend to kDa. Annotate the gels with the molecular weights of the markers.

P5 line 195 - 'Many efforts have been made to improve the ability to express heterologous proteins 195 in E. coli. In this study, we added IPTG to the medium to induce the expression of bcLAP.' This is superfluous and should be removed.

P6 Fig 3 - looks pixellated and stretched. Revise.

Fig 3 - figure legend - (d) - how much IPTG did you add to this series?; are these protein extracts the soluble lysate? You haven't defined how you've made these samples. Please correlate this with the methods explicitly. (d) what are Uu/mL (y-axis)? Correct this. The figure legend lists kD again - please amend to kDa. Annotate the gels with the molecular weights of the markers.

P7 line 227 - replace hydrophobic chromatography with 'hydrophobic interaction chromatography'

Fig 4 - the gel looks odd. Lane 1 is much wider than lane 2. The loading dye bands at the bottom of the gel look like there are potentially 6 lanes that have been loaded in this gel. Replace this gel image by preparing a new SDS-PAGE figure with each purification stage compared - an isolated band with no supporting evidence of how this compared to e.g. the AS precipitated material is not particularly informative.

Fig 5 - revise the figure legend. (a) Optimum temperature. - this doesn't describe the experiment being illustrated here. Please revise the description of each panel to describe the experiment briefly. Particularly the difference between (c) and (d) - I appreciate these are described in detail in the results, but the description here is insufficient for a figure legend.

P9 line 289 - PepA should not have a space.

Table 2 - 'without the addition of an ion' - this cannot be used to infer the absence of a metal ion co-purified with the enzyme from the bacterial lysate. Unless this has been treated with EDTA, you are comparing an unknown bound metal with addition of other metals. Did you EDTA treat the enzyme prior to determining the 100% activity described in the Table legend? Please clarify this in the Table legend. This is highlighted in Table 3 - the fold improvement relative to EDTA treated enzyme would be a more informative measure of the metal ion dependence of this enzyme. This experimental series should be repeated for EDTA treated enzyme.

P9 line 306 - remove space in LAP II

P10 line 333 - alpha-helices, not helixes; and amend the predicted ss content of bcLAP to one decimal place (lines 335-337)

P10 line 342-344 - yes, these are difficult methods; here this hasn't been attempted, so why describe this reasoning? Amend this section to reflect that you have modelled the structure of bcLAP given the availability of a high identity experimental model.

P11 line 358 - reword. It's not a typical sandwich structure. It should be described more accurately as a triple layered structure with an 8-stranded beta-sheet sandwiched between 5-alpha helices on either side of the sheet.

P11 line 371 - amend to reflect the oxidation state 'two Zn2+ ions'

P11 line 373 - the Zn1 binding site doesn't bind to carboxylates; presumably the Zn2+ ion bound at this binding site binds these carboxylates.

P12 line 408 - repetition of (Fig. 7b).

P12 line 409 - brought closer together relative to what? Closer than what? Another model? A structural homologue? Remove or define relativity.

Fig 7 - looks stretched in the x dimension and pixellated again. Was this model generated using PyMOL2.3.2 or was it visualised using PyMOL? Amend. Fig 7c - the Zn2 isn't visible due to the colouring/angle show - remake this figure so these elements are visible.

P13 line 422 - italicise S. olivoreticuli

P13 line 430-446 - comparisons are drawn here between Fig. 7b and Fig. 7d. Please keep the colouring of the residue labels consistent between figures to aid interpretation.

P13 line 463 - double axis - replace with 2-fold axis.

P14 line 470 - 'amino peptidase (SGAK) is docked' sounds like your work - reword to 'has been docked'

Fig. 9 - pixellated and stretched in the x-axis dimension. Review the description of each figure panel. If you are repeating a description throughout, summarise it at the end of the figure legend. Line 478 Green sticks represent zinc - I think these are spheres.

P16 line 493 - Italicise Helicobacter pylori; line 495 - 'fit quite well' is not a scientific description - fit equivalently well perhaps? Can this be supported 

P16 line 503/504 - 'The amino group in the side chain of D-phenylalanine of bestatin and Zn2 did not form a polar contact with Zn2' - should one of these be Zn2+? This doesn't make sense. Also, this should be described as the amino group neighbouring the side chain of D-phenylalanine, not 'in the side chain'.

P18 line 589 and line 593, 595, 608 - italicise bclap

P19 line 623, 625 - what is Tris-HCl buffer?

P19 line 630 - PAGE-SDS - change to SDS-PAGE.

P19 line 624, 626, 647, 650 - rpm is not a universal speed - change to units of xg

P20 line 702 - 'conformation s was...' - amend.

P20 line 708 - kD should be kDa (check this throughout document).

Amend nomenclature of Phyre2, phyre2 and PHYRE2 throughout to be consistent; it is an acronym, so it should be capitalised - and define PHYRE.

P20 line 693 - 'in light of the crystal structure' should read 'based on the crystal structure'

References - 438. Sträter et al? Numbering. 50-57 are further left than the rest of the refs.

Comments on the Quality of English Language

The English in general is excellent. There are a few sentences which I have highlighted in the detailed comments that I think could be amended to clarify their meaning.

Reviewer 2 Report

Comments and Suggestions for Authors

Dear Authors

The present manuscript entitled “Expression, characterization, homology modelling and molecular docking of a novel M17 family leucyl-aminopeptidase from Bacillus cereus CZ” discuss about identification of a new leucyl-aminopeptidase was found in Bacillus cereus which can hydrolyze the peptide bonds in Leu-Trp, Leu-Trp and Ala-Ala-Gly and may be of diverse application. The Manuscript has been planned well and nicely presented. Although there are certain opportunities for further improvement, please find them below.

1.       The introduction is missing some more information regarding background of leucyl-aminopeptidase, mainly a short historical background which may helpful for readers to understand better about evolution of this aminopeptidase over the time.

2.       Figure 2 and 3-please include the marker details (may be an extra picture with band size)to make it clearer.

3.       Discussion can be separate section from results which make it easier for readers.

4.       Line 567- B. cereus CZ was isolated in our laboratory- Please include the sources from where it was isolated and the procedure of isolation of pure culture.

Thank you

Round 2

Reviewer 1 Report

Comments and Suggestions for Authors

You have made substantial changes which I think have improved the article. I disagree with regards to the large lane in the SDS-PAGE image being common in the literature.